# Quantifying institutional-level length of stay variation among hospitalizations for schizophrenia in Ontario between 2014–2021

Andrew Putman[1,2], Joyce Mason[3,4], Phillip Klassen[1,5], David Rudoler[1,2]*

1 Ontario Shores Centre for Mental Health Sciences, Whitby, Ontario, Canada, 2 Faculty of Health Sciences, Ontario Tech University, Oshawa, Ontario, Canada, 3 Centre for Addiction and Mental Health, Toronto, Ontario, Canada, 4 Institute of Health Policy, Management and Evaluation, Dalla Lana School of Public Health, University of Toronto, Toronto, Ontario, Canada, 5 Department of Psychiatry, University of Toronto, Toronto, Ontario, Canada

* David.Rudoler@ontariotechu.ca

## Abstract

Variations in the length and intensity of care delivery from person to person are to be expected, however, such variation can also result from institutional-level factors that may not be directly related to a patient's needs. The objective of this study is to provide a descriptive analysis quantifying the variation in the length of stay (LOS) that is attributable to institutional- and patient-level factors for Ontarians hospitalized with schizophrenia between 2014 and 2021. A retrospective cohort study was conducted using Ontario medical records from >100,000 adult inpatients who had been admitted to an Ontario hospital with a primary diagnosis of schizophrenia between fiscal years 2014 and 2021. The proportion of variation in inpatient LOS that was attributable to institutional-level factors was assessed using log-linear mixed-effects models. Large community, teaching, and specialty mental health hospitals were each modelled separately. These results are presented alongside descriptive analyses for additional context. Average LOS in large community hospitals (mean = 24.2 days, SD = 62.51 days) was lower than specialty mental health hospitals (mean = 85.4 days, SD = 262.9 days) and teaching hospitals (mean = 42.8 days, SD = 137.4 days). The highest proportion of institutional-level variation was seen in large community hospitals (29.3%), with specialty mental health hospitals (19.3%) and teaching hospitals (19.7%) reporting similar proportions of institutional-level variation. This analysis has identified differences in the proportion of inpatient LOS variation attributable to institutional-level factors between types of hospital. A larger percentage of institutional-level variation was seen in large community hospitals however, all hospital types exhibited institutional-level variation. These differences likely result from a combination of governmental and hospital-level factors, which may present an opportunity for policy and clinical interventions. Future research can assess the potential for

**Data availability statement:** The data that support the findings of this study are available from the Ontario Ministry of Health and Long-Term Care: IntelliHEALTH ONTARIO, but restrictions apply to the availability of these data, and so are not publicly available. Data are however available from the authors upon reasonable request and with permission of the Ontario Ministry of Health and Long-Term Care: IntelliHEALTH ONTARIO.

**Funding:** This study was supported by the Citrine Foundation. The funders had no role in study design, data collection and analysis, decision to publish, or preparation of the manuscript.

**Competing interests:** AP, JM, DR declare that they have no competing interests. PK declares having received payment for lectures/presentations from Lundbeck Canada and HLS Therapeutics, Inc.

evidence-based interventions such as bundled care pathways and supportive housing programs in addressing these factors.

## Introduction

Variability in the type, length, and intensity of healthcare delivery is an acceptable and expected part of the delivery of care. However, if variations are caused by organizational factors like culture, budget constraints, or local resources, they can result in inefficiencies in healthcare resource utilization and inequitable variation in health outcomes [1,2]. Previous research indicates that there is significant variation in the delivery of inpatient services for individuals diagnosed with schizophrenia [3–5]. Yet, little is known around the organizational factors that may contribute to this variation. By understanding the level at which variation in care delivery occurs, we can better understand to what extent that variation is being influenced by organizational factors. This improved understanding of the sources of variation can be utilized to identify opportunities for intervention.

Globally, the lifetime prevalence of schizophrenia is around 1%, affecting approximately 80 million people around the world [6,7]. People experiencing schizophrenia-related symptoms typically require intense health care interventions, often including hospital admission. [6] Hospitalizations for schizophrenia and other psychotic disorders currently represent one fifth of all mental health hospitalizations and almost half of all mental health bed days in the province of Ontario, Canada [8]. Additionally, the total number of Ontarians discharged after hospitalization for psychotic disorders has been increasing on average, rising more than 15% between 2016/2017 (18,402 discharges) and 2020/2021 (21,402 discharges). [9] On top of the higher frequency of hospitalizations that people diagnosed with schizophrenia experience, they also typically have longer in-patient length of stays (LOS) than hospitalizations for non-psychotic mental illnesses, both in Canada and globally [3,4,9,10].

An additional layer of complexity in interpreting the length of stay for people hospitalized with schizophrenia is the large person-to-person variability of their inpatient LOS [3,4,10]. Current literature suggests that some of this variation can be attributed to individual-level factors such as clinical characteristics and socio-economic status [4,6,10,11], but there has been less investigation into variation that is related to organizational-level factors [3,10,12]. For example, Chen et al. (2017) found that mean LOS for schizophrenia in specialized psychiatric hospitals in Ontario was more than quadruple the mean LOS for schizophrenia in general hospitals in Ontario between 2005 and 2015. Chen et al. (2017) created separate models of individual-level factors associated with LOS for each hospital setting and reported significant associations between longer LOS and patient demographic characteristics (e.g., age), psychosocial factors (e.g., marital status), and service history (e.g., number of previous admissions).

This paper builds on and updates existing research by using mixed-effects modelling to assess the proportion of LOS variation that is attributable to organizational-level factors. This analysis specifically focuses on the differences in variation at the institutional level by comparing large community hospitals, specialty mental health hospitals, and teaching hospitals.

## Methods

This study used a retrospective cohort design to analyze inpatient hospitalizations for schizophrenia in Ontario between April 1st, 2014, and March 31st, 2022 (fiscal year 2014 through fiscal year 2021). The study time period includes observations from the first two years of the COVID-19 pandemic [13].

### Ethics statement

This study analyzed anonymized Ontario clinical and administrative health record information which was provided by the Ontario Ministry of Health and Long-Term Care through their IntelliHealth Ontario platform [14]. The dataset for this study was primarily extracted from IntelliHealth on May 6th, 2024, with census-linked data (also from Intellihealth) being added on May 15th, 2024. The already anonymized Intellihealth data that was extracted was cleaned for analysis and is securely stored onsite at the Ontario Shores Centre for Mental Health Sciences. This study was approved by the Ontario Shores Centre for Mental Health Sciences Joint Research Ethics Board (JREB # 23–005-D).

### Participants

This study collected information on all Ontario hospital admissions with a primary diagnosis of schizophrenia between April 1st, 2014, and March 31st, 2022. This included instances where an inpatient was not identified as having a primary diagnosis of schizophrenia at the time of admission but was identified as having a primary diagnosis of schizophrenia prior to being discharged from that inpatient stay. Since this analysis is focused on the organization-level influences in LOS variation, in instances where an inter-hospital transfer occurred only the record with the longest LOS from that episode of hospitalization was kept, in the case of ties the most recent record was selected. An inter-hospital transfer within an episode of hospitalization was defined as when a person had been discharged from hospital and then admitted to a different hospital with a primary diagnosis of schizophrenia within 48 hours.

OMHRS records were excluded from the study sample if they did not have a valid health card number, were a forensic admission, or were under 18 years of age at the time of admission. The resulting sample consisted of unique discharges from inpatient care where the patient received a primary diagnosis of schizophrenia during the study's retrospective time period and met the study's exclusion criteria.

### Data source

The data for this study was obtained through IntelliHealth Ontario, a knowledge repository that is managed by the Ontario Ministry of Health [14]. The IntelliHealth portal contains anonymized clinical and administrative data collected from various sectors of the Ontario healthcare system. Specifically, this study used information derived from the Canadian Institute for Health Information (CIHI)'s Ontario Mental Health Reporting System (OMHRS) database for information on inpatient mental health hospital stays, collected using the Resident Assessment Instrument — Mental Health (RAI-MH©) version 2.0 responses [15,16]. The Registered Persons database (RPDB) was used to obtain demographic information, and IntelliHealth's Population Grouper Reporting (PGR) database allowed for postal code linkage of neighbourhood-level sociodemographic variables.

### Primary variables

We captured acute LOS, an integer count of the number of days in which a person required acute inpatient care. Days in which a person is hospitalized are classified as acute care days when their clinicians determine that that person has met the threshold for requiring acute inpatient care [17]. CIHI decision support provides the following guidelines for clinicians determining whether an inpatient's mental health meets any of the following criteria for acute inpatient care on that day:

- *Suffer[s] from sudden and severe psychiatric symptoms; can include patients who are suicidal, have hallucinations, extreme feelings of anxiety, paranoia or depression.*

- *Progressive acute behavioural or neurological difficulties requiring acute clinical or psychiatric care.*

- *Therapeutic pass to inform clinical readiness for discharge.*[17]

We captured alternate level of care (ALC) LOS, an integer count of the number of days in which an inpatient is determined not to have met the threshold for requiring acute inpatient care and are awaiting to be discharged to a more appropriate care setting [17]. We also captured total LOS, a summation of acute and ALC LOS.

We captured the type of hospital using the facility classification of the hospital in the IntelliHealth data: large community hospitals (general hospitals with > 100 inpatient beds, not including hospitals that meet other specialty hospital classifications), specialty mental health hospitals (stand-alone quaternary care hospitals that provide specialized mental health and addiction care), and teaching hospital (hospitals that are directly affiliated with a university medical program, not including specialty mental health hospitals) [18–20]. Hospital types that represented less than 2% of the hospitalizations in this study's sample were collapsed into an `All Other` category when presenting descriptive statistics and were not included in the main regression analysis described below (i.e., amalgamated hospitals, chronic/rehabilitation hospitals, not stated, small hospitals, and specialty children's hospitals).

## Patient-level variables

A primary diagnosis of schizophrenia was determined if the person admitted to inpatient care had an ICD-10-CA or DSM-5 primary diagnosis code of F20 (excluding F20.4), F22, F23, F24, F25, F28, F29, or F53.1 (each DSM-5 diagnosis code has a corresponding ICD-10-CA code); or a DSM-IV primary diagnosis code of 295.xx (10, 20, 30, 40, 60, 70, 80, 90 and 295), 297.1, 297.3, 298.8, or 298.9 [21]. The LOS counts in this analysis include any days which occurred prior to receiving a primary diagnosis of schizophrenia.

Age at admission (in years), education level, and marital status were extracted from the OMHRS dataset. The administrative sex item recoded "other" responses as "intersex" and the income source items from OMHRS were recoded for our dataset into a single categorical item. The Positive Symptom Scale (PSS) – long version is a symptom scale in the RAI-MH that sums the responses from 8 positive symptom-related categorical questions where scores can range from 0 (no positive symptoms present) to 24 (daily presence of positive symptoms for all 8 questions) [16,22]. The self-report RAI-MH items of recent psychiatric admission, recent community MH contact, temporary address at admission, and addiction admission were also extracted from OMHRS as patient-level variables [15,16].

The statistical area classification (SAC) and neighbourhood income quintile of persons in the sample were determined through postal code-linked census data from the PGR database. SACs are Statistics Canada classifications of Canadian census geographic subdivisions into 1 of 3 categories: census metropolitan areas (CMA) – has a population of >100,000 with >50,000 of whom live in the census subdivision's core; census agglomeration area (CA) – has a population of >10,000 living in the census subdivision's core (often a municipality adjacent to a CA); and outside CMA & CA [23,24].

## Handling of missing data

None of the primary variables (type of hospital, acute LOS, ALC LOS, and total LOS), patient age, administrative sex, or the PSS-long variable contained missing data. Data missing from the census-linked neighbourhood income quintile (9.7% across all hospital types) and SAC levels (0.8% across all hospital types) were not assumed to be missing at random due to differences in missingness by hospital type and thus were coded as a discrete `Unknown/Missing` category for each variable. As with the neighbourhood income quintiles and SAC level variables, OMHRS variables with missing data were not assumed to be missing at random and were also coded so that each variable with missing data reported that missing data as a distinct `Unknown/Missing` category.

### Statistical analysis

Overall descriptive statistics (mean, standard deviation (sd), median, interquartile range (IQR)) of the entire sample related to LOS and the type of hospital that reported each discharge were generated. The Coefficient of Variation (CV) for the complete sample and by type of hospital was presented as a standardized measure of how widely each subgroup's LOS values varied around that group's mean LOS, and the distributions of the acute and total LOS lengths were visualized using violin plots [25].

Quantification of the proportion of variation attributable to hospital- and individual-level factors was obtained through mixed-effects modelling. A mixed-effects model was created for each of the 3 types of hospital that frequently treat inpatients for schizophrenia (large community, specialty mental health, and teaching hospitals) using the log of total LOS as its outcome and each hospital representing a random effect. The selection of individual-level explanatory variables was informed by creating a short list of variables that had been previously identified in the literature as being associated with differences in LOS and were individual-level differences [3,10,26–28]. That short list was then further refined by removing explanatory variables that had correlation coefficients of >.7 with one or more explanatory variables. This was done since strong correlation between fixed effects has been shown to have the potential to bias mixed effects model estimates [29]. The individual-level control variables used in each model were age at admission, administrative sex, neighbourhood income quintile, SAC level, PSS-long score at admission, education level, income source, recent psychiatric admission, recent community mental health contact, addictions-related admission, marital status, and temporary address at admission.

Sensitivity analyses as well as stratification by age (≥ 50 years, < 50 years) and sex (females, males) were conducted to further explore the relationship between individual- and hospital-level sources of variation in specific circumstances. The sensitivity analyses include assessing the exclusion ALC days, reducing the number of individual-level explanatory variables, and removing outlier LOS occurrences. Since some patients had repeat hospitalizations during the study period, we also conducted a sensitivity analysis in which we stratified the cohort into patients with unique hospitalizations and those with repeat hospitalizations to determine if the variation in LOS differed for these groups. For the outlier sensitivity analysis, outlier total LOS hospitalizations were determined using the non-parametric upper Tukey fence method (Q3 + 1.5(IQR)) by hospital type [30]. Since we were only interested in estimates of variation in LOS at individual- and hospital-levels, we did not interpret any of the individual-level coefficients or per-hospital random effects.

The analyses were performed in the R statistical computing language [31] version 4.4.1, along with the packages tidyverse [32] version 2.0.0, lme4 [33] version 1.1-35.5, broom.mixed [34] version 0.2.9.5, ggpubr [35] version 0.6.0, and tableone [36] version 0.13.2.

## Results

From the beginning of the 2014 fiscal year through the end of the 2021 fiscal year there were 110,571 discharges with a primary diagnosis of schizophrenia in Ontario. Of those discharge records, we identified 20,748 records which involved at least 1 inter-hospital transfer within a single hospitalization episode. After selecting the record pertaining to the longest LOS portion per hospitalization episode which had been identified as including at least one transfer, 11,721 of those 20,748 transfer-related records were removed. This resulted in a total of 98,850 unique hospitalizations being identified prior to applying the study's exclusion criteria. As seen in Fig 1, the exclusion criteria were then applied, resulting in the removal of 1,313 records for having an invalid health card number, 7,708 records identified as forensic admissions, and 643 records of persons <18 years of age at the time of their admission to inpatient care. This process resulted in the study's sample of 89,186 unique hospitalizations with a primary diagnosis of schizophrenia in Ontario between fiscal years 2014 and 2021.

Overall sample (n = 89,186) characteristics and sample characteristics disambiguated by type of hospital (large community, specialty mental health, teaching, and other) are presented in Table 1. More than half of the overall sample was reported as being male (59.9%) and was on average 40.4 years of age at the time of admission to hospital with a primary

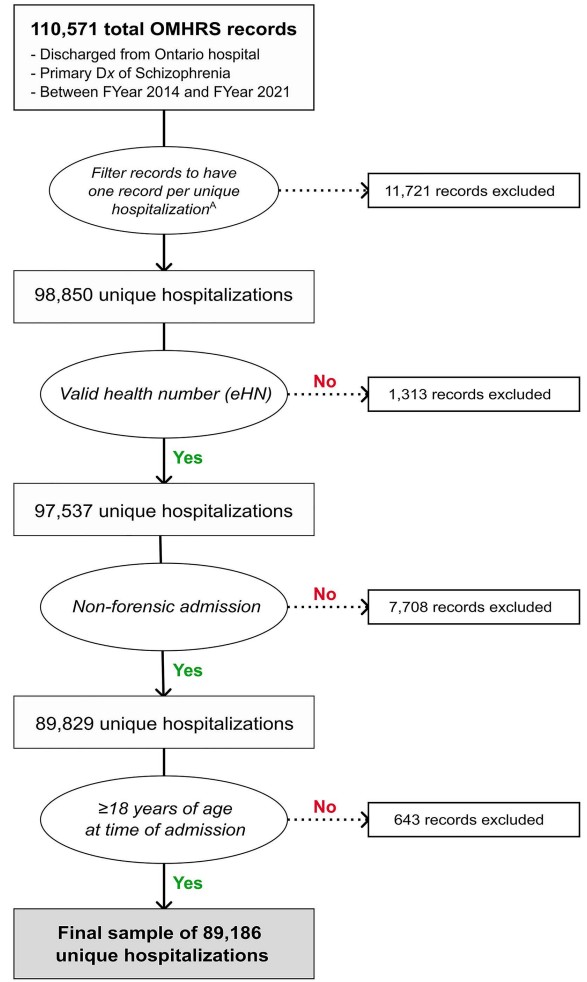

**Fig 1. Flowchart of inclusion and exclusion criteria applied to create the analytic sample.** ^AWhen multiple OMHRS records refer to a single, continuous episode of hospitalization (i.e., having been transferred from one hospital to another), the record with the longest LOS during that episode is kept and attributed to the hospital in which that record occurred.

diagnosis of schizophrenia. Nearly a third (31.9%) of Ontarians admitted to hospital for schizophrenia in our sample were in the lowest neighbourhood income quintile whereas approximately one tenth (10.3%) were in the highest neighbourhood income quintile. This distribution was consistent across hospital types; large community hospitals (lowest quintile = 31.7%, highest quintile = 10.0%), specialty mental health hospitals (lowest quintile = 29.0%, highest quintile = 12.5%), and teaching hospitals (lowest quintile = 34.3%, highest quintile = 9.7%) all reporting between two and three times more people hospitalized with a primary diagnosis of schizophrenia in the lowest income quintile than the highest. Approximately four in five (82.7%) people hospitalized with schizophrenia in our sample lived in census metropolitan areas (CMA), which Statistics Canada defines as an area with at a total population of least 100,000 people and in which at least 50,000 people live in that area's core (city centre) [24]. Teaching hospitals reported an even higher percentage (93.5%) of people from CMAs. Of the RAI-MH variables pulled for analysis, education level has the highest amount of missing data (18.0% overall).

The overall count of unique discharges with a primary diagnosis of schizophrenia in our sample was 10,584 in fiscal year 2014 and rose to 11,893 in 2021. When standardized into a rate of Ontario hospitalizations with a primary diagnosis

**Table 1.** Descriptive characteristics of hospitalizations for schizophrenia in Ontario by type of hospital between 2014 and 2021.

|  | Overall | Type of Hospital | | | |
|---|---|---|---|---|---|
|  |  | Large Community | Specialty MH | Teaching | Other |
| n Observations (%) | 89186 (100.0) | 60824 (68.2) | 10764 (12.1) | 16701 (18.7) | 897 (1.0) |
| n Unique Hospitals (%) | 87 (100.0) | 59 (67.8) | s (s) | 17 (19.5) | s (s) |
| Discharges per [Fiscal] Year (%) |  |  |  |  |  |
| 2014 | 10584 (11.9) | 7404 (12.2) | 1256 (11.7) | 1778 (10.6) | 146 (16.3) |
| 2015 | 10531 (11.8) | 7541 (12.4) | 1190 (11.1) | 1702 (10.2) | 98 (10.9) |
| 2016 | 11037 (12.4) | 7637 (12.6) | 1425 (13.2) | 1862 (11.1) | 113 (12.6) |
| 2017 | 10863 (12.2) | 7423 (12.2) | 1385 (12.9) | 1969 (11.8) | 86 (9.6) |
| 2018 | 10740 (12.0) | 6983 (11.5) | 1443 (13.4) | 2202 (13.2) | 112 (12.5) |
| 2019 | 11868 (13.3) | 7949 (13.1) | 1451 (13.5) | 2356 (14.1) | 112 (12.5) |
| 2020 | 11670 (13.1) | 7871 (12.9) | 1276 (11.9) | 2415 (14.5) | 108 (12.0) |
| 2021 | 11893 (13.3) | 8016 (13.2) | 1338 (12.4) | 2417 (14.5) | 122 (13.6) |
| Acute LOS [in Days] (mean (SD)) | 34.93 (129.33) | 23.35 (50.89) | 78.39 (230.53) | 40.71 (126.71) | 190.77 (703.31) |
| Had ALC Days (%) |  |  |  |  |  |
| Yes | 1912 (2.1) | 727 (1.2) | 635 (5.9) | 433 (2.6) | 117 (13.0) |
| No | 87274 (97.9) | 60097 (98.8) | 10129 (94.1) | 16268 (97.4) | 780 (87.0) |
| ALC LOS [in Days] (mean (SD)) | 2.40 (44.35) | 0.80 (22.07) | 7.01 (61.90) | 2.12 (28.05) | 60.37 (313.40) |
| Total LOS [in Days] (mean (SD)) | 37.33 (156.29) | 24.16 (62.51) | 85.41 (262.87) | 42.82 (137.41) | 251.14 (947.75) |
| Age at Admit (mean (SD)) | 40.45 (15.54) | 39.89 (15.35) | 41.17 (15.88) | 41.71 (15.77) | 46.44 (16.85) |
| Admin. Sex (%) |  |  |  |  |  |
| Female | 35804 (40.1) | 24548 (40.4) | 4066 (37.8) | 6791 (40.7) | s |
| Intersex | 23 (0.0) | 8 (0.0) | 8 (0.1) | 7 (0.0) | s |
| Male | 53359 (59.8) | 36268 (59.6) | 6690 (62.2) | 9903 (59.3) | 498 (55.5) |
| Neighbourhood Income Quintile (%) |  |  |  |  |  |
| 1st - Lowest Income Quintile | 28439 (31.9) | 19308 (31.7) | 3125 (29.0) | 5725 (34.3) | 281 (31.3) |
| 2nd | 16948 (19.0) | 11810 (19.4) | 1933 (18.0) | 3032 (18.2) | 173 (19.3) |
| 3rd | 13900 (15.6) | 9956 (16.4) | 1481 (13.8) | 2335 (14.0) | 128 (14.3) |
| 4th | 12058 (13.5) | 8547 (14.1) | 1473 (13.7) | 1946 (11.7) | 92 (10.3) |
| 5th - Highest Income Quintile | 9184 (10.3) | 6060 (10.0) | 1402 (13.0) | 1614 (9.7) | 108 (12.0) |
| Unknown/Missing | 8657 (9.7) | 5143 (8.5) | 1350 (12.5) | 2049 (12.3) | 115 (12.8) |
| SAC Level (%) |  |  |  |  |  |
| CMA | 73772 (82.7) | 48422 (79.6) | 9276 (86.2) | 15621 (93.5) | 453 (50.5) |
| CA | 8490 (9.5) | 7438 (12.2) | 759 (7.1) | 255 (1.5) | s |
| Outside CMA & CA | 6193 (6.9) | 4565 (7.5) | 600 (5.6) | 624 (3.7) | 404 (45.0) |
| Unknown/Missing | 731 (0.8) | 399 (0.7) | 129 (1.2) | 201 (1.2) | s |
| PSS-Long at Admit (mean (SD)) | 4.09 (4.04) | 4.19 (4.06) | 4.36 (4.14) | 3.53 (3.85) | 4.69 (4.28) |
| Education Level (%) |  |  |  |  |  |
| No Formal Schooling | 2135 (2.4) | 314 (0.5) | 25 (0.2) | 1794 (10.7) | s |
| 8th Grade or Less | 2912 (3.3) | 2074 (3.4) | 349 (3.2) | 441 (2.6) | 48 (5.4) |
| 9th to 11th Grade | 14199 (15.9) | 10260 (16.9) | 1839 (17.1) | 1921 (11.5) | 179 (20.0) |
| Completed High School | 28075 (31.5) | 20645 (33.9) | 2592 (24.1) | 4504 (27.0) | 334 (37.2) |
| Some College or University | 15221 (17.1) | 10788 (17.7) | 2092 (19.4) | 2231 (13.4) | 110 (12.3) |
| Technical or Trade School | 1928 (2.2) | 1313 (2.2) | 206 (1.9) | 404 (2.4) | s |
| Diploma or Bachelor's Degree | 6806 (7.6) | 5027 (8.3) | 418 (3.9) | 1289 (7.7) | 72 (8.0) |
| Graduate Degree or Higher | 1838 (2.1) | 957 (1.6) | 596 (5.5) | 269 (1.6) | 16 (1.8) |
| Unknown/Missing | 16072 (18.0) | 9446 (15.5) | 2647 (24.6) | 3848 (23.0) | 131 (14.6) |

*(Continued)*

**Table 1.** (Continued)

| | | Type of Hospital | | | |
|---|---|---|---|---|---|
| | Overall | Large Community | Specialty MH | Teaching | Other |
| Income Source (%) | | | | | |
| Employment/Pension | 51620 (57.9) | 34791 (57.2) | 6676 (62.0) | 9733 (58.3) | 420 (46.8) |
| Insurance/Social Assistance | 19141 (21.5) | 13621 (22.4) | 1475 (13.7) | 3755 (22.5) | 290 (32.3) |
| Other | 7136 (8.0) | 5183 (8.5) | 709 (6.6) | 1150 (6.9) | 94 (10.5) |
| No Income | 9649 (10.8) | 6794 (11.2) | 1252 (11.6) | 1511 (9.0) | s |
| Unknown/Missing | 1640 (1.8) | 435 (0.7) | 652 (6.1) | 552 (3.3) | s |
| Recent Psychiatric Admission (%) | | | | | |
| 0 Admits in 2 years | 31080 (34.8) | 21626 (35.6) | 2851 (26.5) | 6349 (38.0) | 254 (28.3) |
| 1 or 2 Admits in 2 Years | 33947 (38.1) | 22468 (36.9) | 4681 (43.5) | 6390 (38.3) | 408 (45.5) |
| >2 Admits in 2 Years | 22123 (24.8) | 15867 (26.1) | 2569 (23.9) | 3453 (20.7) | s |
| Unknown/Missing | 2036 (2.3) | 863 (1.4) | 663 (6.2) | 509 (3.0) | s |
| Recent Community MH Contact (%) | | | | | |
| within 30 days | 35926 (40.3) | 22462 (36.9) | 5753 (53.4) | 7147 (42.8) | 564 (62.9) |
| >30 days | 20157 (22.6) | 14501 (23.8) | 2066 (19.2) | 3445 (20.6) | s |
| None | 30815 (34.6) | 22812 (37.5) | 2236 (20.8) | 5581 (33.4) | 186 (20.7) |
| Unknown/Missing | 2288 (2.6) | 1049 (1.7) | 709 (6.6) | 528 (3.2) | s |
| Addictions-Related Admission (%) | | | | | |
| Yes | 17375 (19.5) | 11206 (18.4) | 2629 (24.4) | 3422 (20.5) | 118 (13.2) |
| No | 71811 (80.5) | 49618 (81.6) | 8135 (75.6) | 13279 (79.5) | 779 (86.8) |
| Marital Status (%) | | | | | |
| Married | 10189 (11.4) | 7726 (12.7) | 748 (6.9) | 1635 (9.8) | 80 (8.9) |
| Divorced | 5877 (6.6) | 4045 (6.7) | 715 (6.6) | 1033 (6.2) | 84 (9.4) |
| Partner/Significant Other | 1996 (2.2) | 1490 (2.4) | 165 (1.5) | 313 (1.9) | s |
| Separated | 3081 (3.5) | 2259 (3.7) | 308 (2.9) | 482 (2.9) | 32 (3.6) |
| Single (Never Married) | 65995 (74.0) | 43791 (72.0) | 8618 (80.1) | 12950 (77.5) | 636 (70.9) |
| Widowed | 1750 (2.0) | 1270 (2.1) | 163 (1.5) | 281 (1.7) | 36 (4.0) |
| Unknown/Missing | 298 (0.3) | 243 (0.4) | 47 (0.4) | 7 (0.0) | s |
| Temporary Address at Admission (%) | | | | | |
| Yes | 20841 (23.4) | 13135 (21.6) | 1786 (16.6) | 5792 (34.7) | s |
| No | 66454 (74.5) | 47009 (77.3) | 8244 (76.6) | 10434 (62.5) | 767 (85.5) |
| Unknown/Missing | 1891 (2.1) | 680 (1.1) | 734 (6.8) | 475 (2.8) | s |
| Transfer = no transfer (%) | | | | | |
| Yes | 13053 (14.6) | 6568 (10.8) | 2658 (24.7) | 3464 (20.7) | 363 (40.5) |
| No | 76133 (85.4) | 54256 (89.2) | 8106 (75.3) | 13237 (79.3) | 534 (59.5) |

ALC, alternate level of care; CA, census agglomeration area; CMA, census metropolitan area; LOS, length of stay; MH, mental health [provider]; PSS, Positive Symptoms Scale; *s,* suppressed due to cell size<6.

of schizophrenia per 100,000 people, fiscal year 2014 had a rate of 77.7 unique discharges for schizophrenia per 100,000 Ontarians and in fiscal 2021 that rate was 80.2 unique discharges for schizophrenia per 100,000 Ontarians (which can be seen in Fig 2).

The variation in LOS by hospital type that is seen in Table 1 is expanded upon in Table 2. Large community hospitals had the largest percentage of hospitalizations that were 7 days or less in length (19.1%) and had the lowest percentage of hospitalizations that were 100 days or more in length (2.3%). The standard deviation of acute LOS seen in specialty

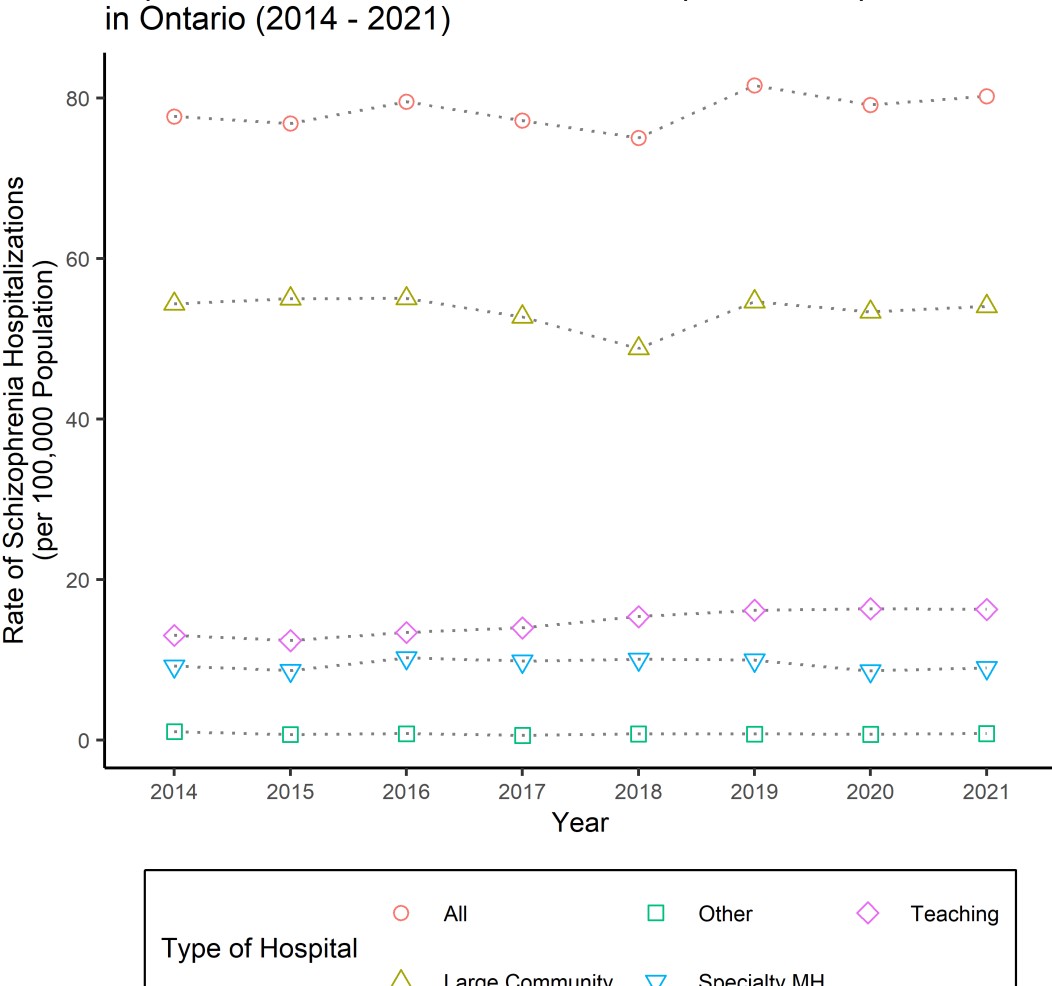

**Fig 2. Population-standardized rate of schizophrenia hospitalizations in Ontario (2014 – 2021).**

mental health hospitalizations for schizophrenia (SD = ±250.53 days, ≈ 7 months and 18 days) is approximately three and a half months longer than schizophrenia hospitalizations in teaching hospitals (SD = ±126.71 days, ≈ 4 months and 5 days), and approximately 6 months longer than schizophrenia hospitalizations in large community hospitals (SD = ±50.89 days, ≈ 1 month and 20 days). After accounting for ALC days, the coefficient of variation for total LOS (total LOS = days acute + days ALC) increased 3% and 4% in teaching hospitals (acute LOS CV = 3.11, total LOS CV = 3.21) and in specialty mental health hospitals (acute LOS CV = 2.94, total LOS CV = 3.08), respectively. The coefficient of variation seen in large community hospitals increased 19% when accounting for ALC days (acute LOS CV = 2.18, total LOS CV = 2.59), but the median and IQR did not change (acute LOS median (IQR) = 15 days (16 days), total LOS median (IQR) = 15 days (16 days)). The skew in LOS observations demonstrated in the differences between mean (SD) and median (IQR) across all hospital types has been visualized in Fig 3.

The results from the mixed-effects analysis (Table 3) demonstrate that approximately 30% of variation in total LOS in large community hospitals (29.30%) was attributable to differences between hospitals, while this was approximately 20%

**Table 2. Variation in LOS for persons hospitalized with a primary diagnosis of schizophrenia in Ontario between 2014 and 2021: By type of hospital and LOS measure.**

| | Type of Hospital | | |
| --- | --- | --- | --- |
| | **Large Community** | **Specialty MH** | **Teaching** |
| **n Hospitalizations (%)** | | | |
| per Type of Hospital | 60824 (100.0) | 10764 (100.0) | 16701 (100.0) |
| Total LOS of <= 7 days | 11612 (19.1) | 1609 (14.9) | 2554 (15.3) |
| Total LOS of>= 100 days | 1413 (2.3) | 2157 (20.0) | 1262 (7.6) |
| **Acute LOS [in days]** | | | |
| mean (SD) | 23.4 (50.89) | 78.4 (230.53) | 40.7 (126.71) |
| median (IQR) | 15 (16) | 31 (64) | 20 (31) |
| CV | 2.18 | 2.94 | 3.11 |
| **Total LOS [in days]** | | | |
| mean (SD) | 24.2 (62.51) | 85.4 (262.87) | 42.8 (137.41) |
| median (IQR) | 15 (16) | 31 (66) | 21 (32) |
| CV | 2.59 | 3.08 | 3.21 |

ALC, alternative level of care; CV, coefficient of variation (SD/mean); LOS, length of stay; MH, mental health.

in specialty mental health (19.34%) and teaching hospitals (19.73%). The combined total amount of observed variation (hospital-level and individual-level) in LOS for inpatients with schizophrenia was highest in specialty mental health hospitals (1.428) and lowest in large community hospitals (.847).

The results seen in Table 3 are consistent with those when only acute LOS days were considered, where the percent of variation attributable to hospital-level differences was 28.53%, 19.51%, and 19.87% for large community, specialty mental health and teaching hospitals respectively. When outliers were removed for each hospital type, the percent variation attributable to hospital-level differences was lower for large community hospitals (8.70%; outlier if total LOS > 49 days) and for teaching hospitals (13.41%; outlier if total LOS > 91 days). Results for specialty mental health hospitals after outlier removal (20.74%; outlier if total LOS > 178 days) remained similar to the findings in the primary analysis. When stratified by age, percent of variation attributable to the hospital level was 10.81% for patients 50 years of age or older in specialty mental health hospitals, approximately 14.5 percentage points lower for patients under 50 years of age in specialty mental health hospitals. This degree of difference by age stratification was not observed in large community or teaching hospitals. Stratification by sex resulted in little difference from the primary results seen in Table 3, with the largest difference in percentage of variation attributable to hospital-level differences being < 2 percentage points (% hospital-level variation in total LOS for only females = 27.58%, vs. 29.30% for the entire sample). Finally, when we limited the analysis to patients with a single hospitalization, the variation attributable to the hospital level increased remained the similar (< 2 percentage points difference) to the primary analysis. When we limited the analysis to patients with multiple hospitalizations, the variation attributable to the hospital level increased from 29.30% to 35.62% for large community hospitals, increased to from 19.73% to 29.09% for teaching hospitals, and remained the similar for specialized mental health hospitals (19.34% to 17.54%). Additional details on the stratification and sensitivity analyses, as well as more detailed model outputs can be found in S1 Appendix.

## Discussion

This study investigated the proportion of hospital-level LOS variation experienced by inpatients with a primary diagnosis of schizophrenia in Ontario hospitals between fiscal years 2014 and 2021. The analysis found evidence of hospital-level variation across all three types of hospital analyzed (large community hospitals, specialty mental health hospitals, and

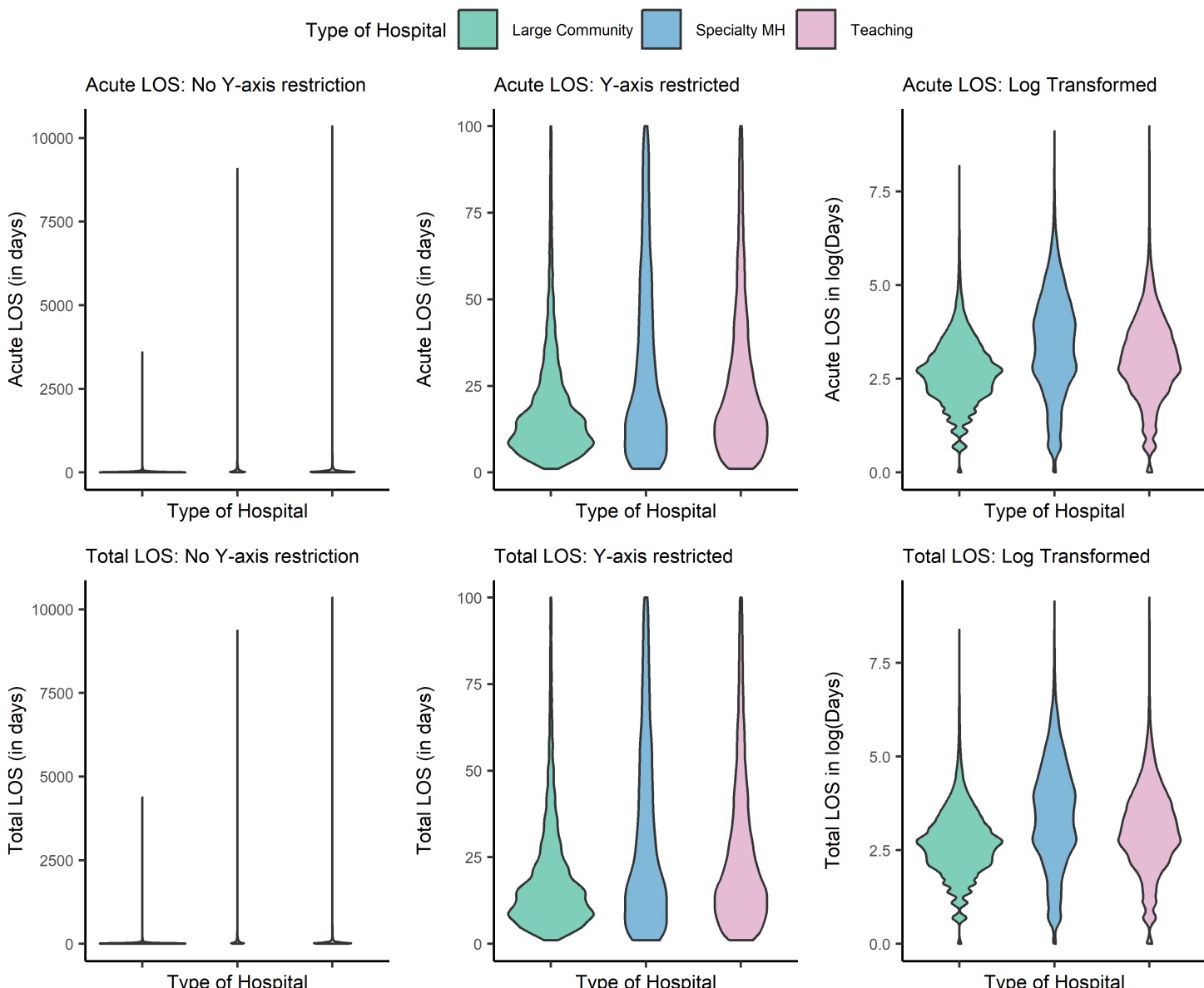

**Fig 3. Violin plot visualizations of schizophrenia LOS distribution in Ontario hospitals between 2014 and 2021 by type of hospital.**

teaching hospitals), with large community hospitals having the highest proportion of variation in LOS attributable to hospital-level differences.

The majority of inpatient stays with a primary diagnosis of schizophrenia during our study period occurred at large community hospitals. These hospitals reported the highest percentage of stays that were a week or less in length and had the fewest stays of 100 or more days. The outlier-removal sensitivity analysis demonstrated that a disproportionate amount of the variation attributable to hospital-level differences in large community hospitals (and to a lesser extent in teaching hospitals) was experienced by a small percentage of inpatients with the longest LOS. Additionally, the sensitivity analysis

**Table 3. Proportion of total LOS variation attributable to individual- and hospital-level factors for persons hospitalized with a primary diagnosis of schizophrenia in Ontario between 2014 and 2021.**

| | Type of Hospital | | |
| --- | --- | --- | --- |
| | Large Community | Speciality MH | Teaching |
| n Hospitalizations | 60824 | 10764 | 16701 |
| n Hospitals | 59 | s | 17 |
| **Mixed-Effects Modelling of log(Total LOS)** | | | |
| Hospital-level Variation | 0.248 | 0.276 | 0.195 |
| Individual-level Variation | 0.599 | 1.152 | 0.794 |
| Proportion of Variation Attributable to Hospital-Level | 29.30% | 19.34% | 19.73% |

LOS, length of stay; MH, mental health; *s,* suppressed due to cell size < 6.

looking at repeat hospitalizations found that individuals who had more than 1 hospitalization within the study period had higher proportions of LOS variation attributable to hospital-level differences in large community and teaching hospitals (but not in specialty mental health hospitals). We found that while specialty mental health hospitals had the widest range of variation (likely due to having higher case complexity), the proportion of variation attributable to hospital-level differences experienced by schizophrenia inpatients at specialty mental health hospitals remained relatively consistent, despite having similar proportions of outliers (≈10% of hospitalizations) as large community and teaching hospitals.

Several studies have attributed unexplained variation in LOS for individuals with severe mental illness to hospital-level factors [5,10,27]. In 2015 Jacobs et al. found considerable unexplained variation in LOS for individuals with severe mental illness in England [10]. They attributed this variation to unobserved factors at the hospital level, such as management culture and efficiency. Similarly, Lee et al. observed that hospital-level factors were associated with LOS for severe mental illness based on data collected in Pennsylvania [5]. Lee et al.'s findings further suggest that unexplained variation in LOS in their study could be the result of differences in treatment philosophies between hospitals [5]. However, much of the research on LOS for severe mental illness has not specifically focused on the impacts of hospital-level factors. For example, in Chen et al.'s 2017 analysis of schizophrenia LOS data in Ontario, hospital-to-hospital variation was controlled for by specifying the admitting hospital as a random effect, but effects of hospital-level differences were not specified in their results [4]. Our results also show considerable variation within and between hospitals, which is in line with the results seen in prior studies that investigated hospital-level factors and LOS variation for inpatients hospitalized with schizophrenia. This study's results further support that a proportion of the variation seen in schizophrenia inpatient LOS are attributable to factors outside of differences in patient characteristics.

The variation in LOS attributable to hospital-level differences seen in our results likely represent a combination of governmental, regional, and hospital-specific factors. These factors may be most amenable to policy and clinical interventions. Variation in care delivery not associated with patient-level factors could be unwarranted variation — variation in care delivery that is not due to patient need or evidence-based practice [1,2]. This unwarranted variation could be explained by provider-level factors (e.g., practice style, practice preferences, responses to outcome uncertainty) as well as practice context and factors related to the practice environment (e.g., differences in available resources, differences in local policy and contextual constraints) [1,37]. As Mercuri et al. explain, unexplained variation in care delivery is often attributed in the literature to one of these factors, but knowing which of the factors is relevant will warrant different policy responses [37]. For example, evidence-based policy interventions such as bundled-care payment models [38] and supportive housing initiatives [39,40] can significantly alter practice context and environment, which has the potential to impact any unwarranted variation that is driven by those factors.

Our study highlights that the variation in LOS differs by hospital type. However, the causes of this variation and the corresponding policy solutions may differ within each hospital type. Hospitalizations for schizophrenia in large community

hospitals may vary due to the heterogeneity of resources, community partnerships, staff composition, and clinical experience and expertise with severe mental illness. In contrast, specialty mental health hospitals may also be heterogeneous but differ in their institutional policies, such as how staff physicians are financially incentivized or whether they discharge patients before securing stable housing. More research is needed to determine the impact of hospital-level factors on the LOS for individuals with schizophrenia, and how these factors may vary by hospital type. This information can help decision-makers in reducing the variability in LOS for this population, which is in line with the Ontario Ministry of Health's *Roadmap to Wellness* [41] plan to improve Ontario's mental health and addictions system.

This analysis has some limitations. First, the administrative data might have miscoded diagnoses, which could result in missed hospitalizations related to schizophrenia and misclassifications among the included cases. Second, the health administrative data has limited sociodemographic variables, which may contribute to variation in LOS and the number of alternate level of care (ALC) days across hospitals. Third, this study utilizes OMHRS data, which reports information about individuals receiving adult inpatient mental health services in Ontario, which can miss hospitalizations for schizophrenia if that person did not receive care that was recorded in OMHRS. This study is descriptive in nature and does not causally assess any specific factor's potential magnitude of effect on the proportion of variation seen in schizophrenia LOS. Future research should further explore which factors may be responsible for the differences seen in the variation of LOS between hospitals and hospital types and to what degree those factors influence the variation in LOS. Additionally, our data does not consistently capture patient-specific information on levels of illness severity (such as treatment refractory schizophrenia) beyond their Positive Symptom Scale – Long (PSS-long) measure. However, there is no reason to believe that differences in illness severity would significantly affect the proportion of LOS variation attributable to differences between hospitals of the same type.

Our study has multiple strengths. To our knowledge, we provide the first quantification of the proportion LOS variation attributable to individual and hospital level differences for patients hospitalized with schizophrenia. Our analysis utilizes data on all individuals admitted to hospitals in Ontario with a primary diagnosis of schizophrenia between fiscal years 2014 and 2021. We build on previous research that focuses on individual-level factors by adding the context of hospital-to-hospital differences. This study serves as a foundation for future investigations into hospital-level factors that influence LOS variation, impacting health system costs, patient health outcomes, and quality of life.

## Conclusion

In this study, we analyzed administrative health data to examine the variation in LOS for individuals hospitalized with schizophrenia across different types of hospitals. We found significant variation in LOS among hospitals of all types, highlighting opportunities to optimize healthcare resource use. Addressing this variation could involve organizational and system-level policies, such as standardizing care delivery and enhancing access to community resources like housing. However, further research is needed to identify the key factors contributing to variation in LOS and to evaluate evidence-based care pathways for this patient population.

## Supporting information

**S1 Appendix. Additional analyses and model coefficients.**
(DOCX)

**S1 Table. Descriptive characteristics stratified by transfer status.**
(XLSX)

## Author contributions

**Conceptualization:** David Rudoler.

**Formal analysis:** Andrew Putman, David Rudoler.

**Methodology:** Andrew Putman, Joyce Mason, David Rudoler.

**Project administration:** Phillip Klassen, David Rudoler.

**Validation:** Andrew Putman.

**Visualization:** Andrew Putman.

**Writing – original draft:** Andrew Putman, David Rudoler.

**Writing – review & editing:** Andrew Putman, Joyce Mason, Phillip Klassen, David Rudoler.

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
