## [Decision Letter · Decision Letter 0]

PMEN-D-24-00561

Quantifying institutional-level length of stay variation among hospitalizations for schizophrenia in Ontario between 2014-2021

PLOS Mental Health

Dear Dr. Rudoler,

Thank you for submitting your manuscript to PLOS Mental Health. After careful consideration, we feel that it has merit but does not fully meet PLOS Mental Health’s publication criteria as it currently stands. Therefore, we invite you to submit a revised version of the manuscript that addresses the points raised during the review process.

Please pay attention to the expert reviewer's specific comments regarding clarity in the methodology as well as terminology use throughout.

We look forward to receiving your revised manuscript.

Kind regards,

Avanti Dey, PhD

Senior Staff Editor

PLOS Mental Health

Journal Requirements:

Additional Editor Comments (if provided):

Reviewers' comments:

Reviewer's Responses to Questions

**Comments to the Author**

1. Does this manuscript meet PLOS Mental Health’s publication criteria?

Reviewer #1: Yes

2. Has the statistical analysis been performed appropriately and rigorously?

Reviewer #1: Yes

3. Have the authors made all data underlying the findings in their manuscript fully available (please refer to the Data Availability Statement at the start of the manuscript PDF file)?

Reviewer #1: Yes

4. Is the manuscript presented in an intelligible fashion and written in standard English?

Reviewer #1: Yes

Reviewer #1: Thank you for the opportunity to review this descriptive analysis looking at variation in length of stay for patients hospitalized with schizophrenia.

I have a few observations I’d like to bring up to explain, why at this time, the manuscript is not ready for publication and major revisions are needed.

1. For the audience’s benefit, it would be useful to list the 5 specialty mental health hospitals captured in the study, the city they are found in, and any unique differences that distinguish their intake process from academic and community hospitals (in general). A 2016 report by the Office of the Auditor General of Ontario lists 4 specialty hospitals (CAMH - Toronto, Ontario Shores - Whitby, The Royal - Ottawa, and Waypoint - Penetanguishene), while an online search also adds St. Joseph’s – London and North Bay Regional (neither making a list of 5): https://www.auditor.on.ca/en/content/annualreports/arreports/en16/v1_312en16.pdf . It is possible that some began operating later in the observation period, and this information as well as other contextual details are important to readers.

2. The authors report that they include two types of primary diagnosis of schizophrenia. The first (and obvious) is where inpatients’ primary diagnosis is defined as schizophrenia as intake (i.e., time at admission); while the second is prior to discharge. One thing I would have liked to see, as part of stratified analysis (and appreciate the inclusion of other such analyses), is models run using this distinction (i.e., comparing hospitalizations where at time of admission schizophrenia was the primary cause compared with at time of discharge). Given the authors retained the longest stay as the outcome for their analysis, it is plausible that state at intake and transfer policies may explain differences in the kinds of patients hospitals treat and were not controlled for with fixed effects…

3. Generally speaking, the authors do a nice job of defining every term they include in their models. I may have missed it, but I would appreciate the inclusion of definitions for ‘recent psychiatric admission’, ‘recent community mental health contact’, and ‘additions-related admission’. Specifically, aside from the information included in appendix 1 (e.g., recent psychiatric admission: 1 or 2 admits in 2 years) I’d like to know what data were used to create these flags, and in the case of ‘addictions-related admission’ clarification on if this measure was flagging additional codes used for the episode of schizophrenia or prior to the event, and if prior – how long in the past. Similarly for the other two definitions, how long a look back was used and ICD codes considered.

4. Early in the results, the authors explain how they identified 100,571 discharge records in the OMHRS, and after removing the shorter stays where a transfer occurred, and a series of cases excluded owing to invalid or missing information, they retained 89,186. I cannot make the numbers work out to this sum when 15,903 were removed as part of the process of retaining a single record per hospitalization episode. This, I feel is the biggest crux of the overall discrepancy issue I observed. While columns in tables added up upon spot checks, the number of hospitalizations retained across the different facility types did not appear to be accurate based on the description provided and the organization of the flow chart.

a. Relatedly, although not mentioned in the model, it is not clear if there were patients with repeat hospitalizations, and if so – were patient repeated measures incorporated in the model (which should be the case if there was a sufficient proportion of hospitalization records that were repeat patients).

5. I understand that given the descriptive nature of the paper, the authors cannot provide explanations for the variations in length of stay observed across facilities. However, given that each mental health hospital (where the variation at the hospital level was largest) is in a unique urban setting, and limited information on how intake and discharge at these sites operate – it is possible that these facilities are servicing very different patients whose differences (as briefly mentioned in the limitations) are not sufficiently accounted for. This is not to discount the authors work, but rather to support the idea that additional discussion on how these 5 sites operate are needed to then further support the idea that ‘more research is needed’ – and specifically how that additional inquiry could be structured.

Minor points

1. The ‘All Other’ category reporting in Table 1 is cut off. While it is available in the second supplementary file – I’m mentioning it here in case other reviewers say they experienced issues reviewing the table.

**Do you want your identity to be public for this peer review?** For information about this choice, including consent withdrawal, please see our Privacy Policy

Reviewer #1: No

---

## [Decision Letter · Decision Letter 1]

PMEN-D-24-00561R1

Quantifying institutional-level length of stay variation among hospitalizations for schizophrenia in Ontario between 2014-2021

PLOS Mental Health

Dear Dr. Rudoler,

Thank you for submitting your manuscript to PLOS Mental Health. After careful consideration, we feel that it has merit but does not fully meet PLOS Mental Health’s publication criteria as it currently stands. Therefore, we invite you to submit a revised version of the manuscript that addresses the points raised during the review process.

EDITOR: Be sure to:

Your work has merit but you just need to address minor concerns raised by the reviewers. Please do not make major revision

Please submit your revised manuscript by **30th June 2025.** If you will need more time than this to complete your revisions, please reply to this message or contact the journal office at mentalhealth@plos.org. Please include the following items when submitting your revised manuscript:

We look forward to receiving your revised manuscript.

Kind regards,

Kizito Omona, PhD

Academic Editor

PLOS Mental Health

Journal Requirements:

Additional Editor Comments (if provided):

Your work has merit but you just need to address minor concerns raised by the reviewers. Please do not make major revision

Reviewers' comments:

Reviewer's Responses to Questions

**Comments to the Author**

Reviewer #2: (No Response)

Reviewer #3: All comments have been addressed

publication criteria?

Reviewer #2: (No Response)

Reviewer #3: Yes

3. Has the statistical analysis been performed appropriately and rigorously?

Reviewer #2: (No Response)

Reviewer #3: Yes

4. Have the authors made all data underlying the findings in their manuscript fully available (please refer to the Data Availability Statement at the start of the manuscript PDF file)?

Reviewer #2: (No Response)

Reviewer #3: Yes

5. Is the manuscript presented in an intelligible fashion and written in standard English?

Reviewer #2: (No Response)

Reviewer #3: Yes

Reviewer #2: Recommendations

1. Methodology

• Clarify Hospital Classification:

• The definition of - speciality mental health hospitals should be explicitly stated (e.g., whether they include forensic units or specific types of care).

• Missing Data Handling:

• Justify the assumption that missing data (e.g., education level, income quintile) are not missing at random. Consider sensitivity analyses (e.g., multiple imputation) to test robustness.

• Transfer Episodes:

• Provide more details on how inter-hospital transfers were identified and handled (e.g., criteria for defining a transfer and impact on LOS calculations).

2. Quality of Analysis

• Outlier Analysis:

• The outlier removal sensitivity analysis is functional, but the Tukey fence method's rationale should be explained.

• Consider alternative methods (e.g., truncation, winsorization) and discuss their impact.

• Repeat Hospitalizations:

• The added sensitivity analysis for repeat hospitalizations is valuable. Expand on how this affects the interpretation of institutional-level variation (e.g., whether certain hospitals disproportionately serve high-frequency users).

3. Citation and Referencing

• Update References:

• Some citations (e.g., Chen et al., 2017) are outdated. Include more recent literature on LOS variation in mental health care.

• Clarify the source of hospital classifications (e.g., Ontario’s Mental Health Act) with specific citations.

Reviewer #3: I congratulate the authors for such a wonderful job and hard work for such informative study. The study has a unique context and concept I understand, yet I would recommend to add 'Acknowledgement' and extract 'Conclusion' after Discussion section at the end. This is recommended for the readability of common people and aspiring future research enthusiasts.

Rest of the comments have been addressed respectively in the first revision. Good luck.

**Do you want your identity to be public for this peer review?** For information about this choice, including consent withdrawal, please see our Privacy Policy

Reviewer #2: No

Reviewer #3: No

---

## [Editor Report · Decision Letter 2]

Quantifying institutional-level length of stay variation among hospitalizations for schizophrenia in Ontario between 2014-2021

PMEN-D-24-00561R2

**Dear Dr. Rudoler,**

We are pleased to inform you that your manuscript 'Quantifying institutional-level length of stay variation among hospitalizations for schizophrenia in Ontario between 2014-2021' has been provisionally accepted for publication in PLOS Mental Health.

Best regards,

Kizito Omona, PhD

Academic Editor

PLOS Mental Health